# Small molecule cognitive enhancer reverses age-related memory decline in mice

Karen Krukowski[1,2]*, Amber Nolan[2,3†], Elma S Frias[1,2†], Morgane Boone[4†], Gonzalo Ureta[5], Katherine Grue[1,2], Maria-Serena Paladini[1,2], Edward Elizarraras[1,2], Luz Delgado[5], Sebastian Bernales[5], Peter Walter[4,6]*, Susanna Rosi[1,2,7,8,9]*

[1]Department of Physical Therapy and Rehabilitation Science, University of California at San Francisco, San Francisco, United States; [2]Brain and Spinal Injury Center, University of California at San Francisco, San Francisco, United States; [3]Department of Pathology, University of California at San Francisco, San Francisco, United States; [4]Biochemistry and Biophysics, University of California at San Francisco, San Francisco, United States; [5]Fundación Ciencia & Vida, Santiago, Chile; [6]Howard Hughes Medical Institute, University of California at San Francisco, San Francisco, United States; [7]Department of Neurological Surgery, University of California at San Francisco, San Francisco, United States; [8]Weill Institute for Neuroscience, University of California at San Francisco, San Francisco, United States; [9]Kavli Institute of Fundamental Neuroscience, University of California at San Francisco, San Francisco, United States

*For correspondence:
karen.krukowski@ucsf.edu (KK);
peter@walterlab.ucsf.edu (PW);
susanna.rosi@ucsf.edu (SR)

†These authors contributed equally to this work

**Abstract** With increased life expectancy, age-associated cognitive decline becomes a growing concern, even in the absence of recognizable neurodegenerative disease. The integrated stress response (ISR) is activated during aging and contributes to age-related brain phenotypes. We demonstrate that treatment with the drug-like small-molecule ISR inhibitor ISRIB reverses ISR activation in the brain, as indicated by decreased levels of activating transcription factor 4 (ATF4) and phosphorylated eukaryotic translation initiation factor eIF2. Furthermore, ISRIB treatment reverses spatial memory deficits and ameliorates working memory in old mice. At the cellular level in the hippocampus, ISR inhibition (i) rescues intrinsic neuronal electrophysiological properties, (ii) restores spine density and (iii) reduces immune profiles, specifically interferon and T cell-mediated responses. Thus, pharmacological interference with the ISR emerges as a promising intervention strategy for combating age-related cognitive decline in otherwise healthy individuals.

## Introduction

"Of the capacities that people hope will remain intact as they get older, perhaps the most treasured is to stay mentally sharp" (*Aging, 2015*).

The impact of age on cognitive performance represents an important quality-of-life and societal concern, especially given our prolonged life expectancy. While often discussed in the context of disease, decreases in executive function as well as learning and memory decrements in older, healthy individuals are common (*Connelly et al., 1991*; *Anderson et al., 1998*; *Kramer et al., 1999*; *Cepeda et al., 2001*). According to the US Department of Commerce, the aging population is estimated by 2050 to reach 83.7 million individuals above 65 years of age in the US; this represents a rapidly growing healthcare and economic concern (*An Aging Nation, 2014*).

Age-related decline in memory has been recapitulated in preclinical studies with old rodents (*Chou et al., 2018*; *Yousef et al., 2019*; *Villeda et al., 2011*; *Castellano et al., 2017*). Specifically, prior studies have identified deficits in spatial memory (*Villeda et al., 2011*; *Villeda et al., 2014*), working and episodic memory (*Yousef et al., 2019*; *Castellano et al., 2017*) and recognition memory (*Cabral-Miranda et al., 2020*), when comparing young, adult mice with older sex-matched animals. The hippocampus is the brain region associated with learning and memory formation and is particularly vulnerable to age-related changes in humans and rodents (*Disterhoft and Oh, 2007*; *McKiernan and Marrone, 2017*; *Oh et al., 2010*; *Rizzo et al., 2014*). Deficits in a number of cellular processes have been suggested as underlying causes based on correlative evidence, including protein synthesis (*Schimanski and Barnes, 2010*), metabolism (*Azzu and Valencak, 2017*), inflammation (*Franceschi et al., 2000*), and immune responses (*Villeda et al., 2011*; *Villeda et al., 2014*; *Baruch et al., 2014*; *Dulken et al., 2019*). While providing a wealth of parameters to assess, by and large the causal molecular underpinnings of age-related memory decline have remained unclear.

The principle that blocking protein synthesis prevents long-term memory storage was discovered many years ago (*Flexner et al., 1962*). With age there is a marked decline of protein synthesis in the brain that correlates with defects in proper protein folding (*Cabral-Miranda et al., 2020*; *López-Otín et al., 2013*; *Ingvar et al., 1985*; *Smith et al., 1995*). Accumulation of misfolded proteins can activate the integrated stress response (ISR) (*Harding et al., 2003*), an evolutionary conserved pathway that decreases protein synthesis. In this way, the ISR may have a causative role in age-related cognitive decline. We previously discovered that interference with the drug-like small-molecule inhibitor (integrated stress response inhibitor, or ISRIB) rescued traumatic brain injury-induced behavioral and cognitive deficits (*Chou et al., 2017*; *Krukowski et al., 2020*; *Costa-Mattioli and Walter, 2020*), suggesting that this pharmacological tool may be useful in testing this notion.

Increasing age leads to structural and functional changes in hippocampal neurons. Specifically, in old animals there is an increase in neuronal hyperpolarization after spiking activity ('afterhyperpolarization', or AHP) that decreases intrinsic neuronal excitability and correlates with memory deficits (*Disterhoft and Oh, 2007*; *McKiernan and Marrone, 2017*; *Oh et al., 2010*; *Rizzo et al., 2014*; *Kaczorowski and Disterhoft, 2009*). Aging also manifests itself with synaptic excitability changes in the hippocampus that correlate with a reduction in the bulbous membrane projections that form the postsynaptic specializations of excitatory synapses, termed dendritic spines (*von Bohlen und Halbach et al., 2006*; *Xu et al., 2018*). Morphological changes in dendritic spine density are critical for spatial learning and memory (*Bloss et al., 2011*; *Yasumatsu et al., 2008*). Whether these age-related neuronal changes can be modified or are linked with ISR activation has yet to be determined.

In addition to neuronal changes, ISR activation can modify immune responses via alterations in cytokine production (*Onat et al., 2019*). Indeed, maladaptive immune responses have been linked with cognitive decline in the old brain (*Yousef et al., 2019*; *Villeda et al., 2011*; *Villeda et al., 2014*; *Baruch et al., 2014*). Initial studies focused on age-associated cytokine responses, including interferon (IFN)-mediated cognitive changes (*Baruch et al., 2014*; *Deczkowska et al., 2017*). Type-I IFN responses can induce age-related phenotypes in rodents. Furthermore, the adaptive immune system (T-cell infiltration into the old brain) can regulate neuronal function via IFN-γ production (*Dulken et al., 2019*), suggesting the possibility that age-induced maladaptive immune responses and the ISR are linked. Here we explore the possibility of ISR inhibition by ISRIB as a potential strategy for modifying age-induced neuronal, immune, and cognitive dysfunction.

## Results

### ISRIB resets the ISR in the brain of old mice

ISR activation leads to global reduction in protein synthesis but also to translational up-regulation of a select subset of mRNAs whose translation is controlled by small upstream open-reading frames in their 5′-UTRs (*Hinnebusch et al., 2016*; *Sonenberg and Hinnebusch, 2009*). One well-studied ISR-upregulated target protein is ATF4 (activating transcription factor 4) (*Chen et al., 2003*; *Pasini et al., 2015*). We recently showed ISRIB administration reversed mild head trauma-induced elevation in ATF4 protein (*Krukowski et al., 2020*). Using the same ISRIB treatment paradigm of daily injections on 3 consecutive days (*Chou et al., 2017*; *Krukowski et al., 2020*), we found

decreased age-associated ATF4 protein levels in mouse brain lysates when compared to vehicle-treated controls during ISRIB administration (*Figure 1—figure supplement 1*). ATF4 levels 18 days after cessation of ISRIB treatment showed persistent reduction in age-induced ATF4 protein levels that were indistinguishable from young mice (*Figure 1A,B*, *Figure 1—figure supplement 2A*).

The key regulatory step in the ISR lies in the phosphorylation of eukaryotic translation initiation factor eIF2 (*26*). Four known kinases can phosphorylate Ser51 in its α-subunit of (eIF2α) to activate the ISR (*Wek, 2018*): HRI (heme-regulated inhibitor), PKR (double-stranded RNA-dependent protein kinase), PERK (PKR-like ER kinase) and GCN2 (General amino acid control nonderepressible 2). Only three of these kinases are known to be expressed in the mammalian brain (PKR, PERK, GCN2). To understand upstream modifiers of age-related ISR activation, we investigated the impact of age and ISRIB administration on the expression of these kinases. We found a modest, but significant increase in activated GCN2 (as indicated by its phosphorylated form p-GCN2) when comparing young and old brain lysates (*Figure 1C*, *Figure 1—figure supplement 2B*). Moreover, when ISRIB was administered weeks prior (*Figure 1A*), GCN2 activation returned to levels comparable to young brains (*Figure 1C*, *Figure 1—figure supplement 2B*). Age and ISRIB did not impact phosphorylation status of PERK and PKR in total brain lysates (*Figure 1D,E*, *Figure 1—figure supplement 2B*). Thus, brief ISRIB administration in the old brain has long-lasting effects on ISR activation.

## Inhibition of the ISR reverses age-induced decline in spatial learning and memory

To assess whether the reduction in ISR activation affects age-related cognitive defects, we tested the capacity for spatial learning and memory in young and old mice in a radial arm water maze (*Chou et al., 2017*; *Alamed et al., 2006*). This particular forced-swim behavior tool measures hippocampal-dependent spatial memory functions in rodents and has been previously used to assess age-related cognitive deficits (*Chou et al., 2018*; *Horowitz et al., 2020*). Animals were trained for 2 days (2 learning blocks/day) to locate a platform hidden under opaque water in one of the eight arms using navigational cues set in the room (*Figure 2A*). We recorded the total number of entries into the non-target arm (errors) before the animal found the escape platform with automated tracking software and used it as a metric of learning. After 2 days of training, young animals averaged one error prior to successfully locating the escape platform, whereas old animals averaged three errors, indicating their reduced learning capacity (*Figure 2—figure supplement 1A*).

We next tested whether pharmacological inhibition of the ISR could modify the age-related spatial learning deficits. ISRIB treatment started the day before the first training day and continued with daily injections throughout the duration of the training (three injections in total; see *Figure 2A*, left). By the end of 2 days of training, ISRIB-treated old animals averaged two errors prior to finding the escape platform, whereas vehicle-treated old animals averaged three, denoting significant learning improvement in the mice that received ISRIB (*Figure 2—figure supplement 1B*). No difference in learning performance was measured in young mice that received the identical treatment paradigm (*Figure 2—figure supplement 1C*), suggesting that ISRIB-induced learning improvement measured in this training regime is age-dependent. These results were confirmed in an independent old animal cohort, in which we tested an additional ISR inhibitor (Cmp-003, a small molecule with improved solubility and pharmacological properties (PCT/US18/65555)), using an identical training/injection paradigm (*Figure 2—figure supplement 1D*). Old animals that received Cmp-003 made significantly fewer errors prior to locating the escape platform than old animals that received vehicle injections, again indicating significant learning improvement.

Spatial memory of the escape location was measured 1 week later by reintroducing the animals into the pool and measuring the number of errors before they located the hidden platform. The animals did not receive any additional treatment during this task. Old mice treated with ISRIB 1 week before made significantly fewer errors compared to matched, vehicle-treated-old male (*Figure 2B*) and female (*Figure 2—figure supplement 2*) mice. Remarkably, the memory performance of old animals treated with ISRIB a week before was comparable to that of young mice (*Figure 2B*). These results demonstrate that brief treatment with ISRIB rescues age-induced spatial learning and memory deficits, cementing a causative role of the ISR on long-term memory dysfunction.

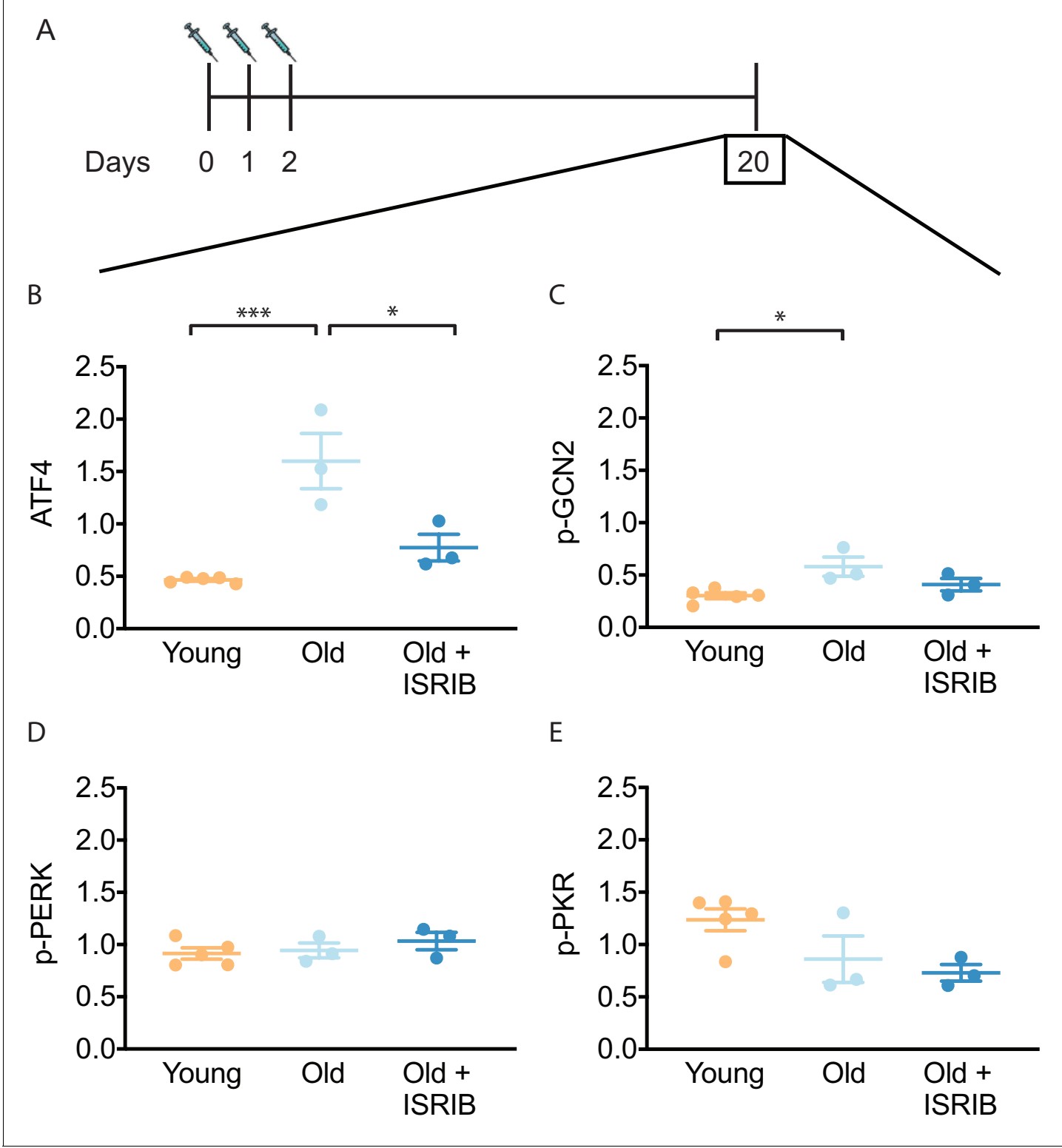

**Figure 1.** ISRIB resets the ISR in the brain of old mice. (**A**) Experimental dosing scheme: ISRIB treatment denoted by syringes (three injections). (**B**) ISRIB treatment reduced ATF4 protein levels chronically 18 days after ISRIB treatment was complete. One-way ANOVA (F = 18.8, p<0.001); with Tukey post-hoc analysis. (**C**) Modest age-induced increases in p-GCN2 when comparing young and old male mice. One-way ANOVA (F = 6.6, p<0.05); with Tukey post-hoc analysis. (**D, E**) Age and ISRIB administration did not impact p-PERK or p-PKR protein levels. Brain lysates of specific protein levels listed

*Figure 1 continued on next page*

*Figure 1 continued*

normalized to actin. Young n = 5, Old = 3, Old + ISRIB = 3. Individual animal values represented by dots; lines depict group mean ± SEM. *p<0.05; ***p<0.001.

The online version of this article includes the following figure supplement(s) for figure 1:

**Figure supplement 1.** ISRIB downregulates ATF4 during administration.

**Figure supplement 2.** ISRIB down-regulates the ISR in the brain of old mice.

## ISRIB administration improves age-induced deficits in working and episodic memory weeks after treatment

Given the long-lasting effect of brief ISRIB treatment on ATF4 protein levels in the brain and on memory function 1 week after drug administration, we next tested the duration of ISRIB effects on age-related cognitive function. On experimental day 20 (18 days post-ISRIB treatment, *Figure 2A*, right), we measured working and episodic memory using a delayed-matching-to-place paradigm (DMP) (*Chou et al., 2017*; *Feng et al., 2017*) in the same animal cohort without additional ISRIB treatment. Previous work has demonstrated that old mice display significant impairments when compared to young mice (*Yousef et al., 2019*; *Castellano et al., 2017*). During DMP animals learned to locate an escape tunnel attached to one of 40 holes in a circular table using visual cues. The escape location was changed daily, forcing the animal to relearn its location. To quantify performance, we used analysis tracking software to measure 'escape latency', reporting the time taken by the mouse to enter the escape tunnel.

Old mice that received ISRIB treatment 18 days earlier displayed significant improvement over the 4-day testing period (*Figure 2C*; Day 20 vs. Day 23). By Day 23, post-treatment animals were locating the escape tunnel on average 20 s faster than the matched-vehicle group (*Figure 2C*). This behavior is indicative of improved working and episodic memory. By contrast, old animals that received vehicle injections did not learn the task (*Figure 2C*; Day 20 vs. Day 23), as previously observed (*Yousef et al., 2019*; *Castellano et al., 2017*). These results demonstrate that ISRIB administration increases cognitive performance in a behavioral paradigm measured weeks after administration.

## ISRIB treatment reverses age-associated changes in hippocampal neuron function

To determine the neurophysiological correlates of ISRIB treatment on cognition, we investigated its effects on hippocampal neuronal function. Utilizing whole cell patch clamping, we recorded intrinsic electrophysiological firing properties and synaptic input in CA1 pyramidal neurons of young and old mice and compared them to those of old mice treated with a single injection of ISRIB the day prior to recording (*Figure 3A*).

We evaluated alterations in intrinsic excitability by measuring action potential shape and frequency properties and passive membrane response properties produced by a series of hyperpolarizing and depolarizing current steps (20 steps from −250 to 700 pA, 250 ms duration). We also assessed the hyperpolarization of the membrane potential following high frequency firing, specifically the AHP following ~50 Hz spiking activity induced with a current step (*Figure 3A*). In agreement with previous reports (*Disterhoft and Oh, 2007*; *McKiernan and Marrone, 2017*; *Oh et al., 2010*; *Rizzo et al., 2014*; *Kaczorowski and Disterhoft, 2009*), old mice displayed a significantly increased AHP amplitude when compared to young mice (*Figure 3B*). ISRIB treatment reversed the age-induced increase in AHP amplitude, rendering the CA1 neuronal response in ISRIB-treated old mice indistinguishable from young mice (*Figure 3B*). We did not find significant differences in other action potential or passive membrane properties between groups (*Figure 3—figure supplement 1*).

We also measured spontaneous excitatory postsynaptic currents (sEPSC), while holding the cell at −75 mV in a voltage clamp. Both the frequency and amplitude of sEPSCs were indistinguishable between groups (*Figure 3—figure supplement 2*). These data support that ISRIB treatment in old animals restores neuronal function to levels comparable to young neurons by affecting intrinsic excitability and specifically reducing the AHP following high-frequency firing.

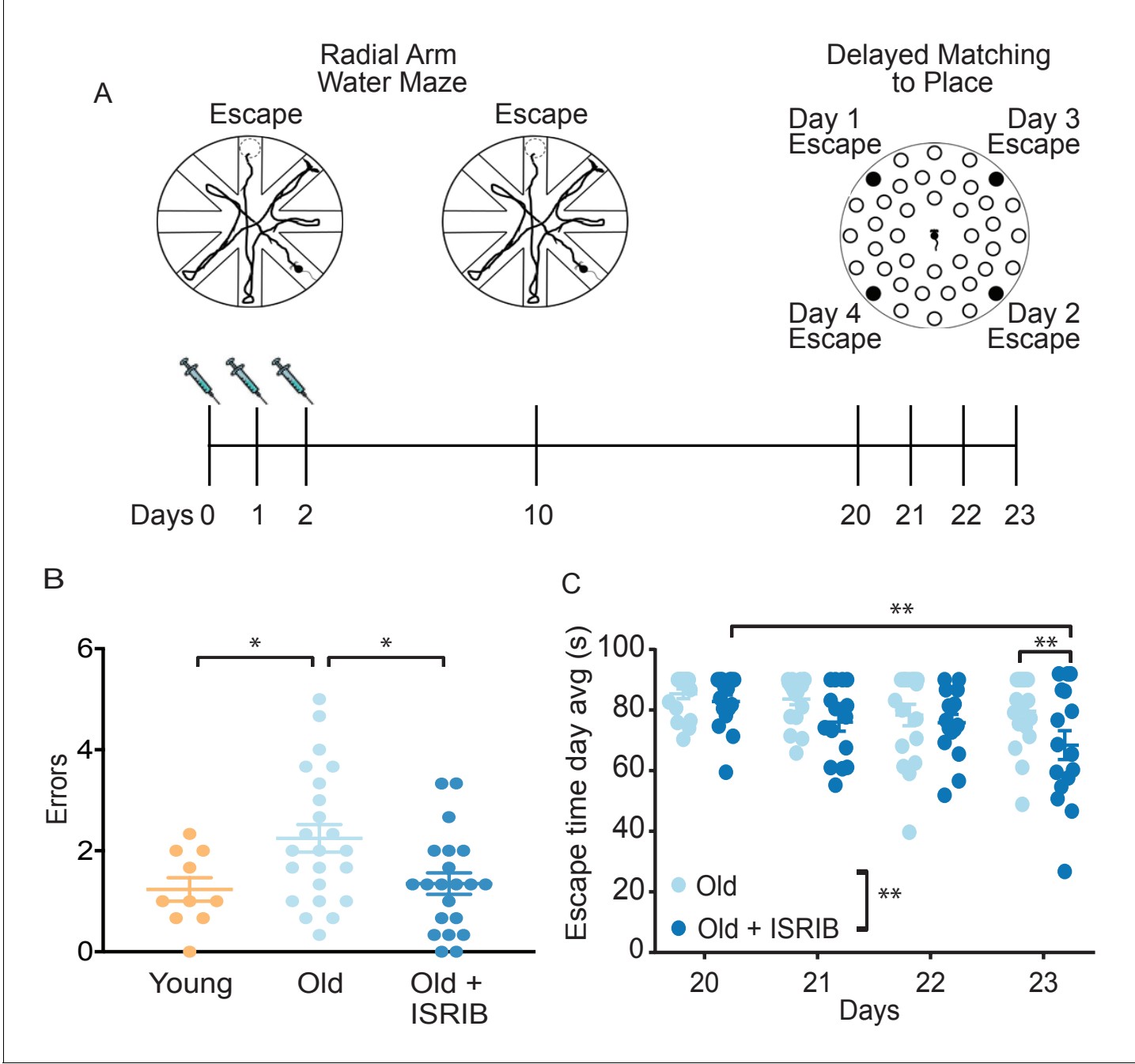

**Figure 2.** Inhibition of the ISR reverses age-induced decline in spatial, working, and episodic memory. (A) Experimental Design: Old (~19 months) animals underwent behavioral analysis in a radial arm water maze (RAWM) and a delayed matching to place paradigm (DMP). ISRIB or vehicle administration (2.5 mg/kg intraperitoneal) occurred daily during the learning phase of RAWM denoted by syringes (days 0–2). (B) ISRIB treatment improved memory 1 week after administration in male rodents. One-way ANOVA (F = 4.8, p<0.05); with Tukey post-hoc analysis. Young n = 10; Old n = 23; Old + ISRIB n = 21. (C) Age-induced deficits in working and episodic learning and memory restored weeks after ISRIB administration. Animals performed the DMP from day 20 to day 23. Average of all trials per group for each day. Days 20, 21 = 4 trials/day. Days 22,23 = 3 trials/day. Two-way repeated measures ANOVA reveals a significant difference between groups p<0.01 (denoted in figure legend) and time effect p<0.01. *p<0.05, **p<0.01. Old n = 18; Old + ISRIB n = 16. Individual animal scores represented by dots; lines depict group mean ± SEM.

The online version of this article includes the following figure supplement(s) for figure 2:

**Figure supplement 1.** ISR inhibitors relieve age-induced deficits in spatial learning.

**Figure supplement 2.** ISRIB reduces age-induced memory deficits in female mice.

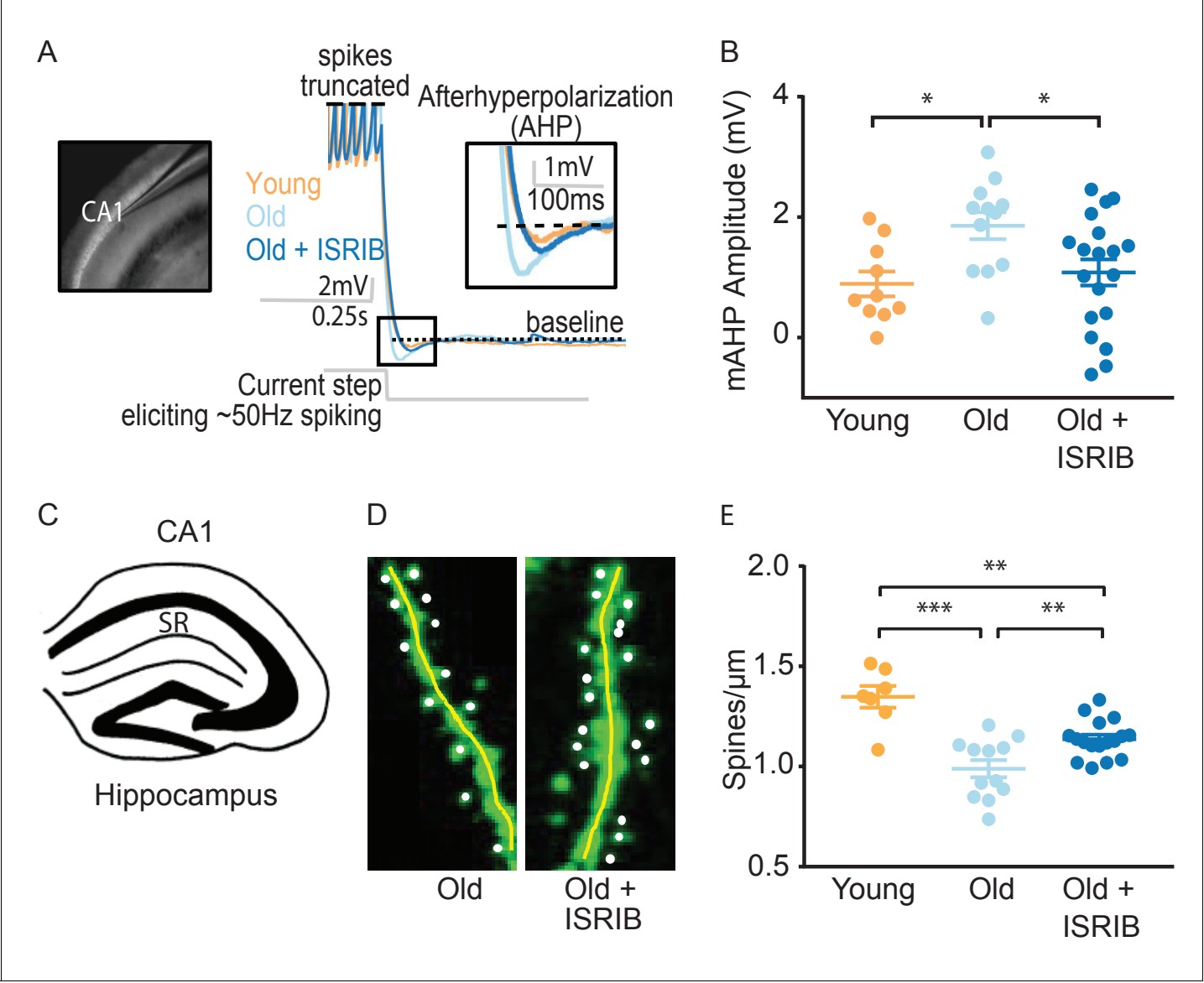

**Figure 3.** ISRIB treatment alleviates age-associated changes in CA1 pyramidal neuron function and structure. (**A**) Left: Image of pipette patched onto CA1 neuron in sagittal slice of hippocampus. Right: Representative traces from hippocampal CA1 pyramidal neurons from old animals treated with either vehicle (light blue) or ISRIB (dark blue) or young animals treated with vehicle (orange) showing the response to a current injection eliciting ~50 Hz spiking activity. Spikes are truncated (dashed line), and the AHP is visualized immediately following cessation of current injection (yellow square) and quantified as the change in voltage from baseline (dotted line). (**B**) Age-induced increases in AHP were measured when comparing young and old animals. ISRIB treatment reversed increased AHP to levels indistinguishable from young animals. Animals were injected with ISRIB (2.5 mg/kg) or vehicle intraperitoneal 1 day prior to recordings. One-way ANOVA (F = 4.461, p<0.05); with Tukey post-hoc analysis. *p<0.05. Each neuron is represented with a symbol; lines indicate the mean ± SEM (Neurons: Young males n = 10 (5 animals); Old males n = 12 (5 animals), Old + ISRIB males n = 19 (7 animals)) with 1–5 neurons recorded per animal. (**C–E**) Spine density was quantified in the CA1 region of the dorsal hippocampus from young and old Thy1-YFP-H mice. (**C**) Diagram of hippocampal region analyzed. SR = stratum radiatum. (**D**) Representative images from Old and Old + ISRIB mice. (**E**) A decrease in dendritic spine density was measured when comparing old mice to young mice. ISRIB treatment significantly increased spine density levels of old mice when compared to vehicle-treated old mice. 63x magnification with a water immersion objective. Young males n = 7 slides (two animals); Old males + Vehicle n = 12 slides (three mice); Old males + ISRIB n = 17 slides (four mice). Individual slide scores (relative to old mice) represented in dots, lines depict group mean ± SEM. One-way ANOVA (F = 18.57, p<0.001) with Tukey post-hoc analysis. **p<0.01; ***p<0.001.

The online version of this article includes the following source data and figure supplement(s) for figure 3:

**Source data 1.** List of electrophysiology reagents.

**Figure supplement 1.** Age and ISRIB treatment do not modify other passive or active intrinsic membrane properties in CA1 pyramidal neurons.

**Figure supplement 2.** Age and ISRIB treatment do not affect spontaneous excitatory post-synaptic currents (sEPSC) in CA1 pyramidal neurons.

## ISRIB treatment reduces dendritic spine loss

To determine if ISRIB might affect age-induced synaptic structural changes, we quantified dendritic spine density after ISRIB treatment in old mice with fluorescently labeled excitatory neurons (marked by a genomically encoded Thy1-YFP fusion protein). The hippocampus of old mice is characterized by a reduction in dendritic spine density that correlates with diminished cognitive output (*von Bohlen und Halbach et al., 2006*; *Xu et al., 2018*). Old Thy1-YFP expressing mice received ISRIB treatment and 2 days of behavioral training as described in *Figure 2A*. At the end of Day 2, we terminated the animals and harvested the brains for quantification of dendritic spine density in the hippocampus (stratum radiatum of CA1) (*Figure 3C*) using confocal microscopy imaging and unbiased analysis (*Figure 3D*). Similar to previous reports, we measured a significant reduction in dendritic spine density in old when compared to young Thy1-YFP mice (*Figure 3E*; *von Bohlen und Halbach et al., 2006*; *Xu et al., 2018*). ISRIB treatment significantly increased spine numbers when compared with age-matched vehicle-treated mice (*Figure 3E*). Taken together these data demonstrate that ISRIB administration improves both neuron structure and function in old mice.

## Age-induced inflammatory tone is reduced following ISRIB treatment

Because it is known that immune dysregulation in the brain increases with age (*Oliveira Pisco et al., 2020*) and correlates with reduced cognitive performance in old animals (*Yousef et al., 2019*; *Villeda et al., 2011*; *Villeda et al., 2014*; *Baruch et al., 2014*), we next investigated immune parameters impacted by ISRIB administration in the old brain. To this end, we first investigated glial cell activation (microglia and astrocytes) in hippocampal sections from of old mice and ISRIB-treated old mice by fluorescent microscopic imaging (during ISRIB administration, *Figure 4—figure supplement 1*). We measured, astrocyte and microglia reactivity as the percent area covered by GFAP and Iba-1. In these analyses, we observed no differences between the old and ISRIB-treated old animals (*Figure 4—figure supplement 1B-G*).

Next, we performed quantitative PCR (qPCR) analyses on hippocampi from young, old, and ISRIB-treated animals on samples taken at the same time point as in the microscopic analysis (*Figure 4—figure supplement 1A*). We measured a panel of inflammatory markers, many of which are known to increase with age (*Oliveira Pisco et al., 2020*), with a particular focus on IFN-related genes as this pathway is implicated in age-related cognitive decline (*Baruch et al., 2014*; *Deczkowska et al., 2017*). Indeed, we found that age increased expression of a number of IFN response pathway genes, *Rtp4, Ifit1, and Gbp10* (*Figure 4A–C*). Importantly, ISRIB administration reduced expression of *Rtp4, Ifit1, and Gbp10* to levels that became indistinguishable from young animals (*Figure 4A–C*). Other inflammatory makers were also increased with age (*Ccl2, Il6*) but not affected by ISRIB treatment, whereas *Cd11b* was increased upon ISRIB administration alone (*Table 1*).

Using these same hippocampal extracts, we next measured T cell responses. Similar to other reports (*Dulken et al., 2019*; *Mrdjen et al., 2018*), we observed a significant increase in T cell marker mRNA expression (*Cd3*) in the hippocampus of old compared to young mice (*Figure 4D*). ISRIB treatment in the old mice reduced the expression of the T cell marker to a level comparable to that observed in young mice (*Figure 4D*). The ISRIB-induced reduction in T cell marker levels was not limited to the brain but extended to the peripheral blood of old animals, with CD8[+] T cell percentages reduced following ISRIB administration (*Figure 4E*). By contrast, we observed no changes in CD4[+] T cell levels (*Figure 4F*).

Given the broad and varied response of immune parameters in response to ISRIB treatment, we next explored possible relationships between behavioral performance and age-related inflammatory tone. T cell marker mRNA expression in the brain positively correlated with cognitive performance: mice with lower T cell marker expression made fewer errors prior to locating the escape platform during memory testing on Day 2 (*Figure 4G*). We also observed significant positive correlations when comparing memory performance on day 2 (errors) with the mRNA levels of multiple IFN response pathway genes (*Ifit1, Rtp4, Gbp10, Gbp5, Oasl1*) and additional inflammatory markers (*Ccl2, Il6*) (*Figure 4G*). These data demonstrate that increased inflammatory marker expression correspond with poorer cognitive performance, even when we did not observed differences between the groups. These studies revealed that ISRIB treatment impacts a broad number of immune

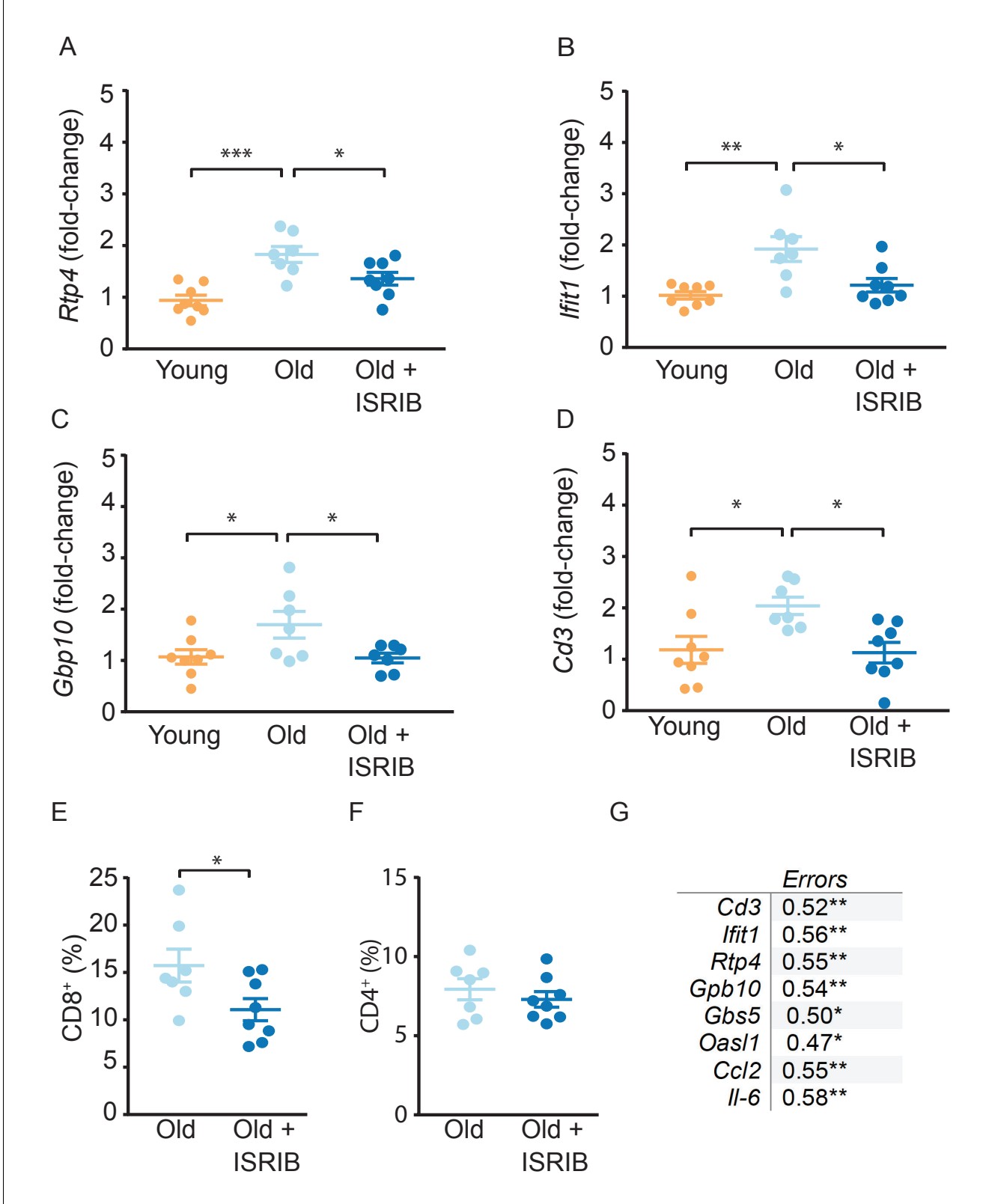

**Figure 4.** Age-induced inflammatory tone is reduced following ISRIB treatment. Inflammatory genes were investigated in the hippocampus of young and old mice by qPCR analysis. (A-C) ISRIB administration reversed age-induced increases in *Rtp4*, *Ifit1*, and *Gbp10*. (A) *Rtp4*, One-way ANOVA (F = 12.23, p<0.001) with a Tukey-post analysis. Young males n = 8; Old males n = 7; Old + ISRIB males n = 8. (B) *Ifit1*, One-way ANOVA (F = 8.8; p<0.01) with a Tukey-post analysis. Young males n = 8; Old males n = 7; Old + ISRIB males n = 8. (C) *Gbp10*, One-way ANOVA (F = F4.2, p<0.05) with a
*Figure 4 continued on next page*

*Figure 4 continued*

Tukey-post analysis. Young males n = 8; Old males n = 7; Old + ISRIB males n = 7. (D) *Cd3* gene-expression (a marker for T cells) changes in the hippocampus of young and old animals were measured by qPCR analysis. *Cd3* was significantly increased with age. ISRIB administration returned *Cd3* expression levels to those comparable to young animals. One-way ANOVA (F = 5.2; p<0.05). Tukey-post hoc analysis. Young males n = 8; Old males n = 7; Old + ISRIB males n = 8. (E, F) Peripheral T cell levels were measured by flow cytometric analysis of whole blood. (E) ISRIB treatment reduced CD8$^+$ T cell percentages (of CD45$^+$ cells) in the peripheral blood. Student t-test. Old males n = 7; Old + ISRIB males n = 8. (F) CD4$^+$ T cell percentages (of CD45$^+$ cells) were not impacted. Individual animal scores represented by dots; lines depict group mean ± SEM. (G) A significant positive correlation was measured between cognitive performance on day 2 of the RAWM (errors) and multiple inflammatory markers (*Cd3, Ifit1, Rtp4, Gbp10, Gbp5, Oasl1, Ccl2, Il-6*). Linear regression was measured by Pearson R correlation, R value denoted with significance. *p<0.05; **p<0.01; ***p<0.001.
The online version of this article includes the following figure supplement(s) for figure 4:

**Figure supplement 1.** ISRIB administration does not impact glial cell activation.

parameters both in the periphery and in the brain reducing the age-related inflammatory tone which strongly correlates with improved cognition.

## ISRIB treatment resets age-related ISR activation

Finally, we investigated the transcriptional expression of ISR mediators and neuronal health markers in the hippocampal lysates used above. We did not detect differences in *Gadd34 (Ppp1r15a), Pkr (Eif2ak2), Bdnf1,* and *Ophn1* mRNA levels with age or ISRIB treatment (*Table 1*). Interestingly however, when analyzed as individual animals, we found a negative correlation between *Ppp1r15a* mRNA and cognitive performance: animals with less *Ppp1r15a* mRNA made more errors prior to locating the escape platform during memory testing (*Figure 5A*). GADD34, the regulatory subunit of one of the two eIF2α phosphatases, acts in a feedback loop as a downstream target of ATF4 (*Brush et al., 2003*; *Connor et al., 2001*; *Novoa et al., 2001*). Induction of GADD34 leads to decreased phosphorylation of eIF2 which counteracts ISR activation. Indeed, ISRIB treatment reduced p-eIF2 levels in total brain lysates (*Figure 5B*; *Figure 5—figure supplement 1*), suggesting that ISRIB may break a feedback loop thereby resetting age-related ISR activation.

**Table 1.** Impact of age and ISRIB on mRNA expression in the hippocampus.
Inflammatory, ISR mediators and neuronal health targets were investigated by qPCR analysis of hippocampal lysates after two ISRIB injections. Columns: (i) mRNA targets (ii) Young group mean ± SEM (iii) Old group mean ± SEM (iv) Old + ISRIB group mean ± SEM (v) ANOVA F value (vi) Significant denotation between groups (vii) n/group.

| TARGET | Young | Old | Old + ISRIB | ANOVA | Btw Groups | n: Yg/Old/Old + ISRIB |
|---|---|---|---|---|---|---|
| *Ccl2* | 1.0 ± 0.1 | 2.3 ± 0.4 | 1.6 ± 0.1 | F = 5.6* | Yg v Old** | 8/7/7 |
| *Cd11b* | 1.0 ± 0.0 | 1.0 ± 0.0 | 1.2 ± 0.0 | F = 6.6** | Yg v Old + ISRIB** Old v Old + ISRIB* | 8/7/8 |
| *Il1-β* | 1.1 ± 0.1 | 1.4 ± 0.1 | 1.7 ± 0.1 | F = 3.6* | Yg v Old + ISRIB* | 8/7/8 |
| Tnf α | 1.0 ± 0.1 | 1.6 ± 0.3 | 2.1 ± 0.4 | F = 2.7 | | 7/7/8 |
| Il-6 | 1.1 ± 0.2 | 1.8 ± 0.1 | 1.8 ± 0.2 | F = 4.1* | Yg v Old* | 8/7/8 |
| Il-10 | 1.0 ± 0.1 | 1.4 ± 0.2 | 1.5 ± 0.2 | F = 1.1 | | 7/7/7 |
| Irf7 | 1.0 ± 0.1 | 1.4 ± 0.1 | 1.2 ± 0.1 | F = 1.5 | | 8/7/8 |
| Ifitm3 | 1.0 ± 0.0 | 1.2 ± 0.1 | 1.0 ± 0.0 | F = 0.8 | | 8/7/8 |
| Isg15 | 1.0 ± 0.0 | 1.3 ± 0.1 | 1.2 ± 0.0 | F = 2.0 | | 8/7/7 |
| Ifi204 | 1.1 ± 0.2 | 1.4 ± 0.2 | 1.5 ± 0.1 | F = 1.2 | | 6/7/7 |
| Gbp5 | 1.0 ± 0.0 | 1.2 ± 0.1 | 1.1 ± 0.0 | F = 2.6 | | 8/7/8 |
| Oasl1 | 1.0 ± 0.0 | 1.8 ± 0.3 | 0.9 ± 0.1 | F = 4.9* | Yg v Old*; Old v Old + ISRIB* | 8/7/8 |
| Ophn1 | 1.0 ± 0.0 | 0.9 ± 0.0 | 1.1 ± 0.0 | F = 1.8 | | 8/7/8 |
| Bdnf | 1.0 ± 0.0 | 0.9 ± 0.0 | 1.0 ± 0.0 | F = 2.7 | | 8/7/8 |
| Gadd34 | 1.0 ± 0.0 | 0.8 ± 0.0 | 0.9 ± 0.0 | F = 1.8 | | 7/7/8 |
| Pkr | 1.0 ± 0.0 | 1.0 ± 0.0 | 1.1 ± 0.0 | F = 1.0 | | 8/7/8 |

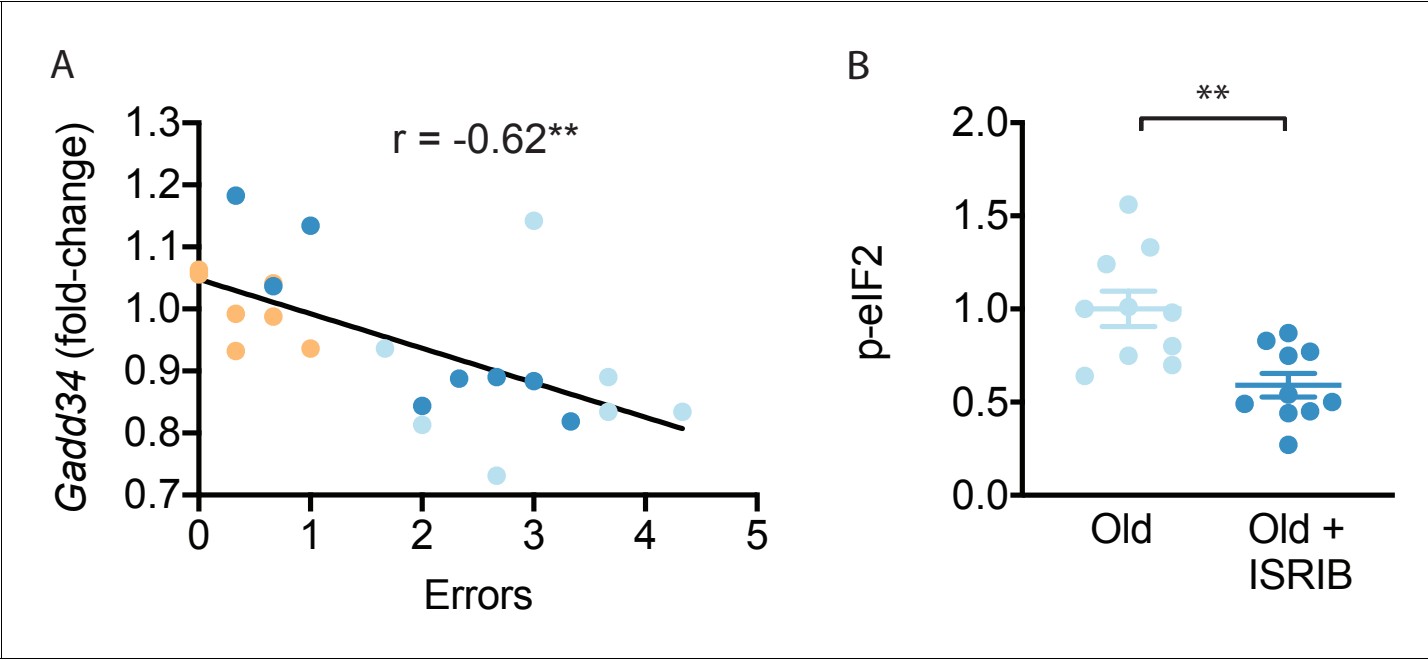

**Figure 5.** ISRIB treatment resets age-related ISR activation. (**A**) A significant negative correlation was measured between cognitive performance on day 2 of the RAWM (errors) and *Ppp1r15a* mRNA expression. Linear regression was measured by Pearson R correlation, R value denoted with significance. (**B**) ISRIB treatment reduced p-eIF2α protein levels. Brain lysates of p-eIF2α protein levels normalized to actin. Old males n = 10; ISRIB males n = 10. Student's t-test. **p<0.01. Data are means ± SEM.

The online version of this article includes the following figure supplement(s) for figure 5:

**Figure supplement 1.** ISRIB treatment breaks age-related ISR activation.

## Discussion

Here, we provide evidence for a direct involvement of the ISR in age-related cognitive decline. Temporary treatment with ISRIB causes a long lasting down-regulation of ATF4 (measured up to 20 days post-treatment). The 'ISR reset' leads to improvement in spatial, working, and episodic memory. At a cellular level, the cognitive enhancement is paralleled by (i) improved intrinsic neuron excitability, (ii) increased dendritic spine density, (iii) reversal of age-induced changes in IFN and T cell responses in the hippocampus and blood, and (iv) reversal of ISR activation. Thus, we identify broad-spectrum anatomical, cellular, and functional changes caused by ISR activation in old animals. If these findings in mice translate into human physiology, they offer hope and a tangible strategy to sustain cognitive ability as we age.

Alterations in proteostasis, the process of protein synthesis, folding and degradation, is a hallmark of aging (*López-Otín et al., 2013*). Accumulation of protein aggregate in the brain is a contributing factor to disease development and progression in several age-related neurodegenerative diseases including Alzheimer's and Parkinson's disease (*Powers et al., 2009*; *Hetz et al., 2019*). The rate of protein synthesis reportedly also declines with age in various regions of the brain (*Schimanski and Barnes, 2010*). Protein synthesis is critical for memory consolidation and the ISR pathway is one of the major nodes for protein synthesis control. Therefore, we investigated how interference with the ISR impacted healthy age-related cognitive decline. ISRIB rapidly enters the brain of old animals and interferes with ISR activation as measured by decreased phosphorylation of eIF2α and ATF4 levels. ISRIB acts downstream of phosphorylation of eIF2α *Sidrauski et al., 2013*; therefore, the decrease in phosphorylation of eIF2α suggests a possible block in the feedback loop that keeps the ISR activated in the old brain; further investigation is needed to understand the exact mechanism. Initial findings suggest this could be through modulation of the kinase GCN2. Here we found that age increased activation of GCN2 (denoted by increased phosphorylation) was attenuated by ISRIB administration. These data indicate that even brief inhibition of the ISR allows the system to 'reset' and dampen the stress response in the old brain.

Along with the ISR, the mammalian target of rapamycin (mTOR), is the other important pathway that cells employ to respond to environmental stress and balance between survival and death. Rapamycin, arguably the most studied and validated small molecule for lifespan expansion, targets mTOR and therefore inhibits mTOR-controlled mRNA translation. However, increased protein production due to mTOR (over)activation can in turn induce the ISR (*Tenkerian et al., 2015*; *Cagnetta et al., 2019*; *Koromilas, 2019*), which serves as a governor keeping protein synthesis in check. Interestingly, despite these two pathways regulating distinct stress responses to translational regulation, it was recently reported that both the ISR and mTOR converge at the level of mRNA translation and both inhibit the translation of the same set of proteins (*Klann et al., 2020*). This complex interplay between mTOR signaling and the ISR could potentially explain how pharmacological treatment with seemingly opposing strategies could both be beneficial.

The reduction in ISR activation corresponded with improved memory performance in old mice weeks after treatment was stopped. Importantly, brief ISRIB treatment improved performance in two different behavioral paradigms (radial arm water maze and the delayed matching to place) previously shown to capture age-related cognitive decline (*Yousef et al., 2019*; *Castellano et al., 2017*; *Villeda et al., 2014*; *Deczkowska et al., 2017*; *Horowitz et al., 2020*; *Smith et al., 2015*). These behavioral paradigms model different aspects of cognitive loss observed during aging in rodents: spatial, working, and episodic memory (*Alamed et al., 2006*; *Feng et al., 2017*). Therefore, our results suggest that ISR inhibition can reverse a broad spectrum of cognitive decline. Notably, spatial learning reversal occurred during ISRIB treatment while the effects on memory were measured weeks after administration and demonstrate the long lasting effects of ISRIB even in the aging brain. Most of the previous studies were performed exclusively in male rodents (*Villeda et al., 2014*; *Deczkowska et al., 2017*; *Horowitz et al., 2020*; *Smith et al., 2015*). Sex-dimorphic cognitive decline has been reported with age in humans (*Casaletto et al., 2019*; *Casaletto et al., 2020*; *Jack et al., 2015*) and rodents (*Davis et al., 2019*; *Davis et al., 2020*). In the current study, we identified cognitive reversal with ISRIB administration in both male and female mice; taking the critical first steps towards studying sex-dimorphic age responses in the ISR. However, we only validated ISRIB-mediated neuronal and immune correlates in male mice. Further work is required to investigate the restorative effects in females.

The neuronal correlates of improved memory function were identified in measures of (i) intrinsic excitability and (ii) dendritic spine number. In CA1 pyramidal neurons, we found that ISRIB treatment reversed the age-associated increase in the afterhyperpolarization (AHP) amplitude following high-frequency (50 Hz) spiking activity to levels found in young mice. AHP is the hyperpolarization of the neuron after an action potential or series of action potentials. The hyperpolarization affects the amount of depolarization needed for subsequent firing thereby influencing intrinsic excitability. An increase of AHP is associated with aging (*Kaczorowski and Disterhoft, 2009*; *Gant et al., 2006*) and impaired learning in young adult rodents (*Oh et al., 2009*). We recapitulated these cellular deficits and we demonstrated that inhibition of the ISR can revert the age-induced increase in intrinsic excitability back to young levels.

Changes in dendritic spine density or structural reorganization of spines are important for synaptic function (*Ultanir et al., 2007*) and cognitive processes such as learning and memory (*Bloss et al., 2011*; *Yasumatsu et al., 2008*; *Frank et al., 2018*; *Lu and Zuo, 2017*). In the aging brain, there is a reduction in hippocampal spine density that has been suggested to be responsible for the poor memory functions (*Cabral-Miranda et al., 2020*; *von Bohlen und Halbach et al., 2006*; *Xu et al., 2018*). A number of age-related molecular processes have been linked to spine loss including disruption in local protein synthesis (*Tolino et al., 2012*). We demonstrated that ISRIB administration alleviated the age-related reductions in hippocampal spine density.

ISRIB down-regulated several immune response pathways in the hippocampus of old animals, including IFN response genes. The importance of this molecule in age-related decline has been demonstrated: Type I IFN responses are increased with age, IFN-β expression in the brain is associated with cognitive decline, and injection of IFN-β expressing viral vectors into an adult mouse can induce an old phenotype (*Baruch et al., 2014*; *Deczkowska et al., 2017*). Here, we found that ISRIB administration reversed age-induced type I IFN responses in the hippocampus suggesting that ISR activation plays a role in this pathway. However, it is still unclear if this is via a direct or indirect effect on inflammatory cells. Several recent publications have identified deleterious roles for T cells in neuronal health or cognitive decline (*Dulken et al., 2019*; *Garber et al., 2019*; *Di Liberto et al., 2018*). Two

recent reports found that persistent T cell infiltration in the brain following viral infection leads to synapse loss (*Garber et al., 2019*; *Di Liberto et al., 2018*). Investigators identified that IFN-γ produced by T-cells-mediated synapse loss (*Garber et al., 2019*; *Di Liberto et al., 2018*). Here, we find that ISRIB treatment impacted T cell levels both in the brain and periphery. The changes in T cells and IFN responses directly correlated with cognitive performance and, again, were restored by ISRIB to the levels observed in young animals. Whether ISRIB reverses cognitive decline by directly modulating one of the measured processes (neuronal intrinsic excitability, dendritic spine number, IFN response, or T cell numbers) or by impacting a number of them remains to be determined.

Aging is an inevitable process for all living creatures, improving our understanding of cellular and molecular processes associated with healthy aging can allow for intervention strategies to modulate cognitive decline. Here, we demonstrate that pharmacological attenuation of the ISR can alleviate age-related neuronal and immune changes potentially resetting age-induced cognitive decline.

## Materials and methods

### Animals

All experiments were conducted in accordance with National Institutes of Health (NIH) Guide for the Care and Use of Laboratory Animals and approved by the Institutional Animal Care and Use Committee of the University of California, San Francisco (Protocol 170302). Male and female C57B6/J wild-type (WT) mice were received from the National Institute of Aging. Thy-1-YFP-H (in C57 background) males were bred and aged in house. Old animals started experimentation at ~19 months of age and young animals 3–6 months of age. Animal shipments were received at least 1 week prior to start of experimentation to allow animals to habituation the new surroundings. Mice were group housed in environmentally controlled conditions with reverse light cycle (12:12 hr light:dark cycle at 21 ± 1°C; ~50% humidity) and provided food and water ad libitum. Behavioral analysis was performed during the dark cycle.

### Drug administration

ISRIB solution was made by dissolving 5 mg ISRIB in 2.5 mLs dimethyl sulfoxide (DMSO) (PanReac AppliChem, 191954.1611). The solution was gently heated in a 40°C water bath and vortexed every 30 s until the solution became clear. Next 1 mL of Tween 80 (Sigma Aldrich, P8074) was added, the solution was gently heated in a 40°C water bath and vortexed every 30 s until the solution became clear. Next, 10 mL of polyethylene glycol 400 (PEG400) (PanReac AppliChem, 142436.1611) solution was added gently heated in a 40°C water bath and vortexed every 30 s until the solution became clear. Finally, 36.5 mL of 5% dextrose (Hospira, RL-3040) was added. The solution was kept at room temperature throughout the experiment. Each solution was used for injections up to 7 day maximum. The vehicle solution consisted of the same chemical composition and concentration (DMSO, Tween 80, PEG400% and 5% dextrose). Stock ISRIB solution was at 0.1 mg/ml and injections were at 2.5 mg/kg. Each animal received an intraperitoneal injection of 2.5x their body weight.

Cmp-003 solution was made by dissolving Cmp-003 (donated by Praxis Biotech) in 50% PEG400 (PanReac AppliChem, 142436.1611) and 50% sterile water. The solution was gently heated in a 40°C water bath and vortexed every 30 s until the solution became clear. Stock Cmp-003 solution was at 0.5 mg/ml and animal injections were at 5.0 mg/kg. Solution was used immediately and made fresh daily.

### Behavioral assessment of cognitive functions

For all assays the experimenter(s) were blinded to therapeutic intervention. Prior to behavioral analysis animals were inspected for gross motor impairments. Animals were inspected for whisker loss, limb immobility (included grip strength) and eye occlusions. If animals displayed *any* of these impairments, they were excluded. Behavioral assessment was recorded and scored using a video tracking and analysis setup (Ethovision XT 8.5, Noldus Information Technology).

### Radial arm water maze

The radial arm water maze (RAWM) was used to test spatial learning and memory in rodents (*Chou et al., 2017*; *Alamed et al., 2006*). The pool is 118.5 cm in diameter with eight arms, each 41

cm in length, and an escape platform. The escape platform is slightly submerged below the water level, so it is not visible to the animals. The pool was filled with water that was rendered opaque by adding white paint (Crayola, 54–2128–053). Visual cues are placed around the room such that they were visible to animals exploring the maze. Animals ran six trials a day during learning and three trials during each memory probe. On both learning and memory days there is a 10 min inter-trial interval. Animals were trained for 2 days and then tested on memory tests 24 hr and 8 days after training. During a trial, animals were placed in a random arm that did not include the escape platform. Animals were allowed 1 min to locate the escape platform. On successfully finding the platform, animals remained there for 10 s before being returned to their warmed, holding cage. On a failed trial, animals were guided to the escape platform and then returned to their holding cage 10 s later. The escape platform location was the same, whereas the start arm varied between trials.

Animals were injected (intraperitoneal) with either vehicle or ISRIB (2.5 mg/kg) starting the day prior to behavior (*Figure 2A*) and after each of the final trials of the learning days (days 1 and 2) for a total of three doses. No injections were given when memory was tested on days 3 and 10. RAWM data were collected through a video tracking and analysis setup (Ethovision XT 8.5, Noldus Information Technology). The program automatically analyzed the number of entries into non-target arms made per trial. Every three trials were averaged into a block to account for large variability in performance; each learning day thus consisted of two blocks, whereas each memory test was one block each. Importantly, in all animal cohorts tested (regardless of age or drug treatment) learning was measured (Significant time effect observed in all Two-way repeated measure ANOVA analysis when groups are analyzed independently).

### Delayed matching to place Barnes maze

Beginning at day 20 animals were tested on DMP using a modified Barnes maze (*Chou et al., 2017*; *Feng et al., 2017*). The maze consisted of a round table 112 cm in diameter with 40 escape holes arranged in three concentric rings consisting of 8, 16, and 16 holes at 20, 35, and 50 cm from the center of the maze, respectively. An escape tunnel was connected to one of the outer holes. Visual cues were placed around the room such that they were visible to animals on the table. Bright overhead lighting and a loud tone (2 KHz, 85 db) were used as aversive stimuli to motivate animals to locate the escape tunnel. The assay was performed for 4 days (days 20–23). The escape tunnel location was moved for each day and animals ran four trials on the first 2 days and three trials on the last 2 days. During a trial, animals were placed onto the center of the table covered by an opaque plastic box so they were not exposed to the environment. After they had been placed on the table for 10 s, the plastic box was removed and the tone started playing, marking the start of the trial. Animals were given 90 s to explore the maze and locate the escape tunnel. When the animals successfully located and entered the escape tunnel, the tone was stopped. If the animals failed to find the escape tunnel after 90 s, they were guided to the escape tunnel before the tone was stopped. Animals remained in the escape tunnel for 10 s before being returned to their home cage. The maze was cleaned with ethanol between each trial. A new escape tunnel was used for each trial. The experimenter was blind to the treatment groups during the behavioral assay. Each trial was recorded using a video tracking and analysis setup (Ethovision XT 8.5, Noldus Information Technology) and the program automatically analyzed the amount of time required to locate the escape tunnel. Animal improvement was calculated by Day 20 escape latency – Day 23 escape latency.

### Tissue collection

All mice were lethally overdosed using a mixture of ketamine (10 mg/ml) and xylaxine (1 mg/ml). Once animals were completely anesthetized, blood was extracted by cardiac puncture and animals were perfused with 1X phosphate buffer solution, pH 7.4 (Gibco, Big Cabin, OK, −70011–044) until the livers were clear (~1–2 min). For Western blot analysis following phosphate buffered solution (PBS), the whole brain (regions dissected discussed below) was rapidly removed and snap frozen on dry ice and stored at −80°C until processing.

### Western blot analysis

Animals received all three ISRIB injections and were terminated 20 hr after the third injection (as described above). Frozen brain lysates or hippocampi isolates were then homogenized with a T 10

basic ULTRA-TURRAX (IKa) in ice-cold buffer lysis (Cell Signaling 9803) and protease and phosphatase inhibitors (Roche). Lysates were sonicated for 3 min and centrifuged at 13,000 rpm for 20 min at 4°C. Protein concentration in supernatants was determined using BCA Protein Assay Kit (Pierce). Equal amount of proteins was loaded on SDS-PAGE gels. Proteins were transferred onto 0.2 µm PVDF membranes (BioRad) and probed with primary antibodies diluted in Tris-buffered saline supplemented with 0.1% Tween 20% and 3% bovine serum albumin.

ATF4 (11815) (Cell Signaling), p-GCN2 (Abcam Cat No ab-75836), p-PERK (Cell Signaling Cat No 3179), p-PKR (Abcam Cat No ab-32036), and p-eIF2 (Cell Signaling, Cat No 3597) and β-actin (Sigma-Aldrich) antibodies were used as primary antibodies. HRP-conjugated secondary antibodies (Rockland) were employed to detect immune-reactive bands using enhanced chemiluminescence (ECL Western Blotting Substrate, Pierce) according to the manufacturer instructions. Quantification of protein bands was done by densitometry using ImageJ software.

ATF4, p-GCN2, p-PERK, p-PKR, and p-eIF2 levels were normalized to β-actin expression and fold-change was calculated as the levels relative to the expression in vehicle-treated derived samples, which corresponds to 1.

## Electrophysiology

Sagittal brain slices (250 µm) including the hippocampus were prepared from old mice (~19 mo) treated with either vehicle or ISRIB or young mice (~3 mo), treated with vehicle, 12–18 hr prior (n = 5, 7, and five per group, respectively). Mice were anesthetized with Euthasol (0.1 ml / 25 g, Virbac, Fort Worth, TX, NDC-051311-050-01), and transcardially perfused with an ice-cold sucrose cutting solution containing (in mM): 210 sucrose, 1.25 $NaH_2PO_4$, 25 $NaHCO_3$, 2.5 KCl, 0.5 $CaCl_2$, 7 $MgCl_2$, seven dextrose, 1.3 ascorbic acid, three sodium pyruvate (bubbled with 95% $O_2$ − 5% $CO_2$, pH 7.4) (see *Figure 3—source data 1* for reagent information). Mice were then decapitated and the brain was isolated in the same sucrose solution and cut on a slicing vibratome (Leica, VT1200S, Leica Microsystems, Wetzlar, Germany). Slices were incubated in a holding solution (composed of (in mM): 125 NaCl, 2.5 KCl, 1.25 $NaH_2PO_4$, 25 $NaHCO_3$, 2 $CaCl_2$, 2 $MgCl_2$, 10 dextrose, 1.3 ascorbic acid, three sodium pyruvate, bubbled with 95% $O_2$ − 5% $CO_2$, pH 7.4) at 36°C for 30 min and then at room temperature for at least 30 min until recording.

Whole cell recordings were obtained from these slices in a submersion chamber with a heated (32–34°C) artificial cerebrospinal fluid (aCSF) containing (in mM): 125 NaCl, 3 KCl, 1.25 $NaH_2PO_4$, 25 $NaHCO_3$, 2 $CaCl_2$, 1 $MgCl_2$, 10 dextrose (bubbled with 95% $O_2$ - 5% $CO_2$, pH 7.4). Patch pipettes (3–6 MΩ) were manufactured from filamented borosilicate glass capillaries (Sutter Instruments, Novato, CA, BF100-58-10) and filled with an intracellular solution containing (in mM): 135 KGluconate, 5 KCl, 10 HEPES, 4 NaCl, 4 MgATP, 0.3 $Na_3GTP$, 7 2K-phosphcreatine, and 1–2% biocytin. CA1 pyramidal neurons were identified using infrared microscopy with a 40x water-immersion objective (Olympus, Burlingame, CA). Recordings were made using a Multiclamp 700B (Molecular Devices, San Jose, CA) amplifier, which was connected to the computer with a Digidata 1440A ADC (Molecular Devices, San Jose, CA), and recorded at a sampling rate of 20 kHz with pClamp software (Molecular Devices, San Jose, CA). We did not correct for the junction potential, but access resistance and pipette capacitance were appropriately compensated before each recording.

The passive membrane and active action potential spiking characteristics were assessed by injection of a series of hyperpolarizing and depolarizing current steps with a duration of 250 ms from −250 pA to 700 nA (in increments of 50 pA). The resting membrane potential was the measured voltage of the cell 5 min after obtaining whole cell configuration without current injection. A holding current was then applied to maintain the neuron at −67 +/- 2 mV before/after current injections. The input resistance was determined from the steady-state voltage reached during the −50 pA current injection. The membrane time constant was the time required to reach 63% of the maximum change in voltage for the −50 pA current injection. Action potential parameters including the half width, threshold, and amplitude were quantified from the first action potential elicited. Action potential times were detected by recording the time at which the positive slope of the membrane potential crossed 0 mV. From the action potential times, the instantaneous frequency for each action potential was determined (1/inter spike interval). The maximum firing frequency was the highest frequency of firing identified throughout all current injections. Action potential rate as a function of current injection was examined by plotting the first instantaneous action potential frequency versus current injection amplitude. The F/I slope was then determined from the best linear fit of the positive values of

this plot. The action potential or spike threshold was defined as the voltage at which the third derivative of V (d3V/dt) was maximal just prior to the action potential peak. The action potential (AP) amplitude was calculated by measuring the voltage difference between the peak voltage of the action potential and the spike threshold. The half-width of the action potential was determined as the duration of the action potential at half the amplitude. The adaptation index of each cell was the ratio of the last over the first instantaneous firing frequency, calculated at 250 pA above the current step that first elicited spiking. The afterhyperpolarization (AHP) was calculated as the change in voltage from baseline (measured as the mean voltage over a 100 ms interval 600 ms after termination of a current injection that first elicited at least 12 spikes corresponding to a firing frequency of ~50 Hz) compared to immediately after cessation of current injection (the minimum voltage reached in the first 175 ms immediately after cessation of current injection). Cells were excluded from analysis if excessive synaptic input was noted during recording of the mAHP or if the cell did not fire at least 12 spikes during current injections.

To measure the spontaneous excitatory postsynaptic currents (sEPSCs), cells were recorded in voltage clamp at a holding potential of −75 mV for 4 min, a holding potential that should have little inhibitory components given the reversal potential of chloride with these solutions. Analysis of sEPSCs was performed using a template matching algorithm in ClampFit 10.7 (Molecular Devices, San Jose, CA). The template was created using recordings from multiple pyramidal cells and included several hundred synaptic events. Access resistance (Ra) was monitored during recordings, and recordings were terminated if Ra exceeded 30 megaohms. Only stable recordings (<50 pA baseline change) with a low baseline noise (<8 pA root mean square) were included. The first 250 synaptic events or all the events measured in the 4-min interval from each cell were included for analysis.

## Fluorescent spine imaging preparation

For fluorescent spine analysis, following PBS animals were perfused with ice-cold 4% paraformaldehyde, pH 7.5 (PFA, Sigma Aldrich, St. Louis, MO, 441244) and fixed for 4–24 hr followed by sucrose (Fisher Science Education, Nazareth, PA, S25590A) protection (15% to 30%). Brains were embedded with 30% sucrose/Optimal Cutting Temperature Compound (Tissue Tek, Radnor, PA, 4583) mixture on dry ice and stored at −80℃. Brains were sectioned into 20 µm slides using a Leica cryostat (Leica Microsystems, Wetzlar, Germany) and mounted on slides (ThermoFisher Scientific, South San Francisco, CA). Slides were brought to room temperature (20℃) prior to use. Tissues were fixed using ProLong Gold (Invitrogen, Carlsbad, CA, P36930) and a standard slide cover sealed with nail polish.

## Spine density quantification

For spine density quantification, whole brains from young and old male Thy1-YFP-H transgenic line were used. Three to six images separated by 60–140 µm in the dorsal hippocampus were imaged per animal and used for dendritic spine density analysis. 9.3 µm z-stack images were acquired on a Zeiss Laser-Scanning Confocal microscope (Zeiss LSM 780 NLO FLIM) at the HDFCCC Laboratory for Cell Analysis Shared Resource Facility. 63x magnification with a water immersion objective. All protrusions from the dendrites were manually counted as spines regardless of morphology. Two individuals (blinded to age and treatment) analyzed a total length of at least 3200 µm of dendrites from each animal using NIH FIJI analysis software (v1.52n). Individual dendritic spine was calculated as density per micron and graphed relative to old mice.

## qPCR analysis

Hippocampus samples, of approximately the same size per animal were process as previously described (*Krukowski et al., 2018a*; *Krukowski et al., 2018b*). Relative gene expression was determined using the $2^{-\Delta\Delta Ct}$ method and normalized using GAPDH. Primers used were the following:

> *Cd3*: Fw 5' TGACCTCATCGCAACTCTGCTC-3' Rev 5' TCAGCAGTGCTTGAACCTCAGC-3'
> *Ifit1*: Fw 5' CTGAGATGTCACTTCACATGGAA-3' Rev 5' GTGCATCCCCAATGGGTTCT-3'
> *Rtp4*: Fw 5' TGGGAGCAGACATTTCAAGAAC-3', Rev 5'ACCTGAGCAGAGGTCCAACTT-3'
> *Gbp10*: Fw 5' GGAGGCTCAAGAGAAAAGTCACA-3', Rev 5' AAGGAAAGCCTTTTGATCCTTCAGC-3'
> *Ccl2*: Fw 5' GCTGACCCCAAGAAGGAATG-3' Rev 5' GTGCTTGAGGTGGTTGTGGA-3'

*Il1β*: Fw 5' TGTAATGAAAGACGGCACACC-3' Rev 5' TCTTCTTTGGGTATTGCTTGG-3'
*Tnfα*: Fw 5' TGCCTATGTCTCAGCCTCTTC-3' Rev 5' GAGGCCATTTGGGAACTTCT-3'
*Il-6*: Fw 5' TACCACTTCACAAGTCGGAGGC-3' Rev 5' CTGCAAGTGCATCATCGTTGTTC-3'
*Irf7*: Fw 5'- GAGACTGGCTATTGGGGGAG-3' Rev 5'- GACCGAAATGCTTCCAGGG-3'
*Ifitm3*: Fw 5'- CCCCCAAACTACGAAAGAATCA-3' Rev 5'- ACCATCTTCCGATCCCTAGAC-3'
*Isg15*: Fw 5'- GGTGTCCGTGACTAACTCCAT-3' Rev 5'- TGGAAAGGGTAAGACCGTCCT-3'
*Ifi204*: Fw 5'- AGCTGATTCTGGATTGGGCA-3' Rev 5'- GTGATGTTTCTCCTGTTACTTCTGA-3'
*Eif2ak2 (Pkr)*: Fw 5'- CTGGTTCAGGTGTCACCAAAC-3' Rev 5'- ACAACGCTAGAGGATG
TTCCG-3'
*Cd11b*: Fw 5'- CTGAGACTGGAGGCAACCAT- 3' Rev 5' GATATCTCCTTCGCGCAGAC-3'
*Il-10*: Fw 5'- GCCAAGCCTTATCGGAAATG- 3' Rev 5' CACCCAGGGAATTCAAATGC −3'
*Bdnf*: Fw 5'- GGCTGACACTTTTGAGCACGT - 3' Rev 5' CTCCAAAGGCACTTGACTGCTG −3'
*Ophn1*: Fw 5'- CTTCCAGGACAGCCAACCATTG - 3' Rev 5' CTTAGCACCTGGCTTCTGTTCC
−3'
*Gbs5*: Fw 5'- CTGAACTCAGATTTTGTGCAGGA - 3' Rev 5' CATCGACATAAGTCAGCACCAG
−3'
*Oasl1*: Fw 5'- CAGGAGCTGTACGGCTTCC - 3' Rev 5' CCTACCTTGAGTACCTTGAGCAC −3'
*Ppp1r15a*: Fw 5'- GGCGGCTCAGATTGTTCAAAGC - 3' Rev 5' CCAGACAGCAAGGAAA
TGGACTG −3'
*Gapdh*: Fw 5' AAATGGTGAAGGTCGGTGTG-3' Rev 5' TGAAGGGGTCGTTGATGG-3'

## Flow cytometric analysis

To assess circulating cell populations peripheral blood was collected by cardiac puncture and transferred into an EDTA collection tube. Blood was aliquoted into flow cytometry staining tubes and stained with surface antibodies for 30–60 min at room temperature (*Krukowski et al., 2018c*). Surface antibodies included anti-CD45 (FITC-conjugated; BD Biosciences), Ly-6G (PE-conjugated; BD Biosciences), CD8 (PE-Cy7-conjugated; BD Biosciences), CD4 (APC-conjugated; BD B), and CD11b (APC-Cy7; BD Biosciences). Leukocyte subpopulations were identified as follows: Forward and side scatter was used to exclude debris and doublet populations. Specific T- cell populations were identified as follows: CD4 T-cell subsets were CD4$^+$, CD45$^+$, Ly-6G$^-$, CD8$^-$, CD11b$^-$. CD8 T-cell subsets were CD8$^+$, CD45$^+$, Ly-6G$^-$, CD4$^-$,CD11b$^-$. After surface antibody staining, red blood cells were lysed with RBC lysis (BD Biosciences). Cell population gating occurred as previously described (*Krukowski et al., 2018c*). (Data were collected on an LSRII (BD)) and analyzed with Flowjo software (v10, Tree Star Inc).

## Statistical analysis

Results were analyzed using Prism software or IBM SPSS Statistics. Individual animals. Individual animal scores represented by dots, lines depict group mean and SEM. Student t-test, one-way ANOVA, two-way repeated measures ANOVA and Pearson R correlations were used (individual statistical tool and post-hoc analysis denoted in Figure Legends). p values of < 0.05 considered as significant. Group outliers were determined (GraphPad Software Outlier Test-Grubb's test) and excluded from analysis. At most a single animal was excluded from each experimental cohort.

The numbers of mice used was sufficient to result in statistically significant differences using standard power calculations with alpha = 0.05 and a power of 0.8. We used an online tool to calculate power and samples size based on experience with the respective tests, variability of the assays and inter-individual differences within groups. All experiments were randomized and blinded by an independent researcher prior to treatment.

Biological replicates are measurements of biologically distinct animals/tissues/cells used to measure biological differences. Technical replicates are repeated measurements of the same animals/tissues/cells.

## Acknowledgements

This work was supported by the generous support of the Rogers Family (to SR and PW), the UCSF Weill Innovation Award (to SR and PW), the NIH/National Institute on Aging Grant R01AG056770 (to SR), the NRSA post-doctoral fellowship from the NIA F32AG054126 (to KK), the National Institute for General Medicine (NIGMS) Initiative for Maximizing Student Development (R25GM056847) and

the National Science Foundation (NSF) Graduate Fellowship Program (To ESF), the UCSF Clinical and National Center for Advanced Translational Sciences at NIH (UCSF-CTSI Grant Number TL1 TR001871) and the NIH/NINDS (K08NS114170) (To AN), the Programa de Apoyo a Centros con Financiamiento Basal AFB 170004 (to SB at "Fundacion Ciencia & Vida"). PW is an Investigator of the Howard Hughes Medical Institute.

We thank Dr. Vikaas Sohal for providing equipment for electrophysiological recordings and advice on analysis. We thank Dr. Spyros Darmanis and Rene Sit from the Chan Zuckerberg Biohub for their assistance with analysis.

## Additional information

### Competing interests

Gonzalo Ureta, Luz Delgado: works at Fundacion Ciencia & Vida and receives partial funding from Praxis Biotech. Sebastian Bernales: is an employee of Praxis Biotech and Fundacion Ciencia & Vida, and receives partial funding from Praxis Biotech. Peter Walter: is an inventor on US Patent 9708247 held by the Regents of the University of California that describes ISRIB and its analogs. Rights to the invention have been licensed by UCSF to Calico. PW is an equity owner and consultant for Praxis Biotech LLC. The other authors declare that no competing interests exist.

### Funding

| Funder | Grant reference number | Author |
| --- | --- | --- |
| National Institute on Aging | F32AG054126 | Karen Krukowski |
| National Institutes of Health | R01AG056770 | Susanna Rosi |
| National Center for Advancing Translational Sciences | TL1 TR001871 | Amber Nolan |
| National Institute of Neurological Disorders and Stroke | K08NS114170 | Amber Nolan |
| Howard Hughes Medical Institute | | Peter Walter |
| UCSF Foundation | Weill Innovation Award | Peter Walter Susanna Rosi |
| National Institute of General Medical Sciences | R25GM056847 | Elma S Frias |
| National School Foundation | Graduate Fellowship Program | Elma S Frias |
| Programa de Apoyo a Centros con Financiamiento Basal | AFB 170004 | Sebastian Bernales |

The funders had no role in study design, data collection and interpretation, or the decision to submit the work for publication.

### Author contributions

Karen Krukowski, Conceptualization, Data curation, Methodology, Writing - original draft, Writing - review and editing; Amber Nolan, Data curation, Funding acquisition, Methodology, Writing - review and editing; Elma S Frias, Morgane Boone, Gonzalo Ureta, Katherine Grue, Maria-Serena Paladini, Edward Elizarraras, Luz Delgado, Data curation, Methodology, Writing - review and editing; Sebastian Bernales, Conceptualization, Resources, Writing - review and editing; Peter Walter, Susanna Rosi, Conceptualization, Supervision, Funding acquisition, Writing - review and editing

### Author ORCIDs

Karen Krukowski (iD) https://orcid.org/0000-0003-0281-8917
Morgane Boone (iD) http://orcid.org/0000-0002-7807-5542

Peter Walter [ORCID] https://orcid.org/0000-0002-6849-708X
Susanna Rosi [ORCID] https://orcid.org/0000-0002-9269-3638

### Ethics

Animal experimentation: This study was performed in strict accordance with the recommendations in the Guide for the Care and Use of Laboratory Animals of the National Institutes of Health. All of the animals were handled according to approved institutional animal care and use committee (IACUC) protocols of the University of California, San Francisco.(Protocol 170302).

### Decision letter and Author response

Decision letter https://doi.org/10.7554/eLife.62048.sa1
Author response https://doi.org/10.7554/eLife.62048.sa2

## Additional files

### Supplementary files

• Transparent reporting form

### Data availability

All data generated or analysed during this study are included in the manuscript and supporting files.

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
