## [Decision Letter]

**Acceptance summary:**

This is a well-written and timely manuscript suggesting novel compounds targeting ISR to enhance learning and memory in old mice. The small molecule ISR inhibitor ISRIB was previously shown to improve memory after traumatic brain injury in a rodent model. Here they show that a single three dose regimen given to old male mice (19 months) rescues the age-related increase in ATF4 and peIF2 protein levels, improves learning and memory in two visuospatial testing paradigms, and restores more youthful neurophysiological phenotypes and dendritic spine numbers. Further, age-related increases in IFN-related genes were reduced, as was the proportion of peripheral CD8 T cells.

**Decision letter after peer review:**

Thank you for submitting your article "Small molecule cognitive enhancer reverses age-related memory decline in mice" for consideration by *eLife*. Your article has been reviewed by three peer reviewers, and the evaluation has been overseen by a Reviewing Editor and Jessica Tyler as the Senior Editor. The following individuals involved in review of your submission have agreed to reveal their identity: Catherine Kaczorowski (Reviewer #1); Alessandro Bitto (Reviewer #3).

The reviewers have discussed the reviews with one another and the Reviewing Editor has drafted this decision to help you prepare a revised submission.

Summary:

This is a well-written and timely manuscript suggesting novel compounds targeting ISR to enhance learning and memory in old mice. The small molecule ISR inhibitor ISRIB was previously shown to improve memory after traumatic brain injury in a rodent model. Here they show that a single three-dose regimen given to old male mice (19 months) rescues the age-related increase in ATF4 and p-eIF2 protein levels, improves learning and memory in two visuospatial testing paradigms, and restores more youthful neurophysiological phenotypes and dendritic spine numbers. Further, age-related increases in IFN-related genes were reduced, as was the proportion of peripheral CD8 T cells.

Essential revisions:

1) The group mean for the crucial data in Figure 2B (Old) is oddly positioned. A rough estimation of the visible data points suggests the mean is closer to 2 than 3. 16 data points are below the shown mean, only 7 above, and the visible distribution is not such that a handful of extreme values could pull the mean so far. The data for this panel should be confirmed, the mean calculation double-checked, and the statistical tests repeated. This is arguably the single most critical piece of data underlying the core conclusion about memory function.

2) The bar graphs need to be redrawn to avoid the white space and the y axis should be adjusted accordingly – especially in Figure 1.

---

## [Author Response]

Essential revisions:1) The group mean for the crucial data in Figure 2B (Old) is oddly positioned. A rough estimation of the visible data points suggests the mean is closer to 2 than 3. 16 data points are below the shown mean, only 7 above, and the visible distribution is not such that a handful of extreme values could pull the mean so far. The data for this panel should be confirmed, the mean calculation double-checked, and the statistical tests repeated. This is arguably the single most critical piece of data underlying the core conclusion about memory function.

Thank you for noting this, we carefully double-checked our data, graphs, means and the statistics reported in the manuscript. As you described the group mean for the Old group is between 2-3, actual value = 2.24. The group mean for the ISRIB group is 1.34. The stats have been rerun and the graph has been recreated; the axis were adjusted so this is more visually apparent to the reader.

2) The bar graphs need to be redrawn to avoid the white space and the y axis should be adjusted accordingly – especially in Figure 1.

In the original submission for clarity of the reader, we maintained the same y axis values for all graphs within each figure. We respectfully disagree with this revision and feel the figure has become misleading with the changes suggested by the reviewer (‘avoidance of white space’). We believe the original Figure 1 best conveys the findings.